# GRAPH INFORMATION BOTTLENECK FOR SUBGRAPH RECOGNITION

**Junchi Yu**[1,2,3]*, **Tingyang Xu**[3], **Yu Rong**[3], **Yatao Bian**[3], **Junzhou Huang**[3], **Ran He**[1,2,4]†

[1]NLPR&CRIPAC, Institute of Automation, Chinese Academy of Sciences, China
[2]University of Chinese Academy of Sciences, China
[3]Tencent AI LAB, China
[4]Center for Excellence in Brain Science and Intelligence Technology, CAS, China
`yujunchi2019@ia.ac.cn, tingyangxu@tencent.com, yu.rong@hotmail.com`
`yatao.bian@gmail.com, jzhuang@uta.edu, rhe@nlpr.ia.ac.cn`

## ABSTRACT

Given the input graph and its label/property, several key problems of graph learning, such as finding interpretable subgraphs, graph denoising and graph compression, can be attributed to the fundamental problem of recognizing a subgraph of the original one. This subgraph shall be as informative as possible, yet contains less redundant and noisy structure. This problem setting is closely related to the well-known information bottleneck (IB) principle, which, however, has less been studied for the irregular graph data and graph neural networks (GNNs). In this paper, we propose a framework of Graph Information Bottleneck (GIB) for the subgraph recognition problem in deep graph learning. Under this framework, one can recognize the maximally informative yet compressive subgraph, named *IB-subgraph*. However, the GIB objective is notoriously hard to optimize, mostly due to the intractability of the mutual information of irregular graph data and the unstable optimization process. In order to tackle these challenges, we propose: i) a GIB objective based-on a mutual information estimator for the irregular graph data; ii) a bi-level optimization scheme to maximize the GIB objective; iii) a connectivity loss to stabilize the optimization process. We evaluate the properties of the IB-subgraph in three application scenarios: improvement of graph classification, graph interpretation and graph denoising. Extensive experiments demonstrate that the information-theoretic IB-subgraph enjoys superior graph properties.

## 1 INTRODUCTION

Classifying the underlying labels or properties of graphs is a fundamental problem in deep graph learning with applications across many fields, such as biochemistry and social network analysis. However, real world graphs are likely to contain redundant even noisy information (Franceschi et al., 2019; Yu et al., 2019), which poses a huge negative impact for graph classification. This triggers an interesting problem of recognizing an informative yet compressed subgraph from the original graph. For example, in drug discovery, when viewing molecules as graphs with atoms as nodes and chemical bonds as edges, biochemists are interested in identifying the subgraphs that mostly represent certain properties of the molecules, namely the functional groups (Jin et al., 2020b; Gilmer et al., 2017). In graph representation learning, the predictive subgraph highlights the vital substructure for graph classification, and provides an alternative way for yielding graph representation besides mean/sum aggregation (Kipf & Welling, 2017; Velickovic et al., 2017; Xu et al., 2019) and pooling aggregation (Ying et al., 2018; Lee et al., 2019; Bianchi et al., 2020). In graph attack and defense, it is vital to purify a perturbed graph and mine the robust structures for classification (Jin et al., 2020a).

Recently, the mechanism of self-attentive aggregation (Li et al., 2019) somehow discovers a vital substructure at node level with a well-selected threshold. However, this method only identifies isolated important nodes but ignores the topological information at subgraph level. Consequently, it

---
*This work was done when Junchi Yu was a research intern at Tencent AI LAB.
†Corresponding Author

leads to a novel challenge as subgraph recognition: *How can we recognize a compressed subgraph with minimum information loss in terms of predicting the graph labels/properties?*

Recalling the above challenge, there is a similar problem setting in information theory called information bottleneck (IB) principle (Tishby et al., 1999), which aims to juice out a compressed data from the original data that keeps most predictive information of labels or properties. Enhanced with deep learning, IB can learn informative representation from regular data in the fields of computer vision (Peng et al., 2019; Alemi et al., 2017; Luo et al., 2019), reinforcement learning (Goyal et al., 2019; Igl et al., 2019) and natural language precessing (Wang et al., 2020). However, current IB methods, like VIB (Alemi et al., 2017), is still incapable for irregular graph data. It is still challenging for IB to compress irregular graph data, like a subgraph from an original graph, with a minimum information loss.

Hence, we advance the IB principle for irregular graph data to resolve the proposed subgraph recognition problem, which leads to a novel principle, Graph Information Bottleneck (GIB). Different from prior researches in IB that aims to learn an optimal representation of the input data in the hidden space, GIB directly reveals the vital substructure in the subgraph level. We first i) leverage the mutual information estimator from Deep Variational Information Bottleneck (VIB) (Alemi et al., 2017) for irregular graph data as the GIB objective. However, VIB is intractable to compute the mutual information without knowing the distribution forms, especially on graph data. To tackle this issue, ii) we adopt a bi-level optimization scheme to maximize the GIB objective. Meanwhile, the continuous relaxation that we adopt to approach the discrete selection of subgraph will lead to unstable optimization process. To further stabilize the training process and encourage a compact subgraph, iii) we propose a novel connectivity loss to assist GIB to effectively discover the maximally informative but compressed subgraph, which is defined as *IB-subgraph*. By optimizing the above GIB objective and connectivity loss, one can recognize the IB-subgraph without any explicit subgraph annotation. On the other hand, iv) GIB is model-agnostic and can be easily plugged into various Graph Neural Networks (GNNs).

We evaluate the properties of the IB-subgraph in three application scenarios: improvement of graph classification, graph interpretation, and graph denoising. Extensive experiments on both synthetic and real world datasets demonstrate that the information-theoretic IB-subgraph enjoys superior graph properties compared to the subgraphs found by SOTA baselines.

## 2 RELATED WORK

**Graph Classification.** In recent literature, there is a surge of interest in adopting graph neural networks (GNN) in graph classification. The core idea is to aggregate all the node information for graph representation. A typical implementation is the mean/sum aggregation (Kipf & Welling, 2017; Xu et al., 2019), which is to average or sum up the node embeddings. An alternative way is to leverage the hierarchical structure of graphs, which leads to the pooling aggregation (Ying et al., 2018; Zhang et al., 2018; Lee et al., 2019; Bianchi et al., 2020). When tackling with the redundant and noisy graphs, these approaches will likely to result in sub-optimal graph representation. Recently, InfoGraph (Sun et al., 2019) maximize the mutual information between graph representations and multi-level local representations to obtain more informative global representations.

**Information Bottleneck.** Information bottleneck (IB), originally proposed for signal processing, attempts to find a short code of the input signal but preserve maximum information of the code (Tishby et al., 1999). (Alemi et al., 2017) firstly bridges the gap between IB and the deep learning, and proposed variational information bottleneck (VIB). Nowadays, IB and VIB have been wildly employed in computer vision (Peng et al., 2019; Luo et al., 2019), reinforcement learning (Goyal et al., 2019; Igl et al., 2019), natural language processing (Wang et al., 2020) and speech and acoustics (Qian et al., 2020) due to the capability of learning compact and meaningful representations. However, IB is less researched on irregular graphs due to the intractability of mutual information.

**Subgraph Discovery.** Traditional subgraph discovery includes dense subgraph discovery and frequent subgraph mining. Dense subgraph discovery aims to find the subgraph with the highest density (e.g. the number of edges over the number of nodes (Fang et al., 2019; Gionis & Tsourakakis, 2015)). Frequent subgraph mining is to look for the most common substructure among graphs (Yan & Yan, 2002; Ketkar et al., 2005; Zaki, 2005). At node-level, researchers discover the vital substructure

via the attention mechanism (Velickovic et al., 2017; Lee et al., 2019; Knyazev et al., 2019). Ying et al. (2019) further identifies the important computational graph for node classification. Alsentzer et al. (2020) discovers subgraph representations with specific topology given subgraph-level annotation. Recently, it is popular to select a neighborhood subgraph of a central node to do message passing in node representation learning. DropEdge (Rong et al., 2020) relieves the over-smoothing phenomenon in deep GCNs by randomly dropping a portion of edges in graph data. Similar to DropEdge, DropNode (Chen et al., 2018; Hamilton et al., 2017; Huang et al., 2018) principle is also widely adopted in node representation learning. FastGCN (Chen et al., 2018) and ASGCN (Huang et al., 2018) accelerate GCN training via node sampling. GraphSAGE (Hamilton et al., 2017) leverages neighborhood sampling for inductive node representation learning. NeuralSparse (Zheng et al., 2020) select Top-K (K is a hyper-parameter) task-relevant 1-hop neighbors of a central node for robust node classification. Similarly, researchers discover the vital substructure at node level via the attention mechanism (Velickovic et al., 2017; Lee et al., 2019; Knyazev et al., 2019).

## 3 NOTATIONS AND PRELIMINARIES

Let $\{(\mathcal{G}_1, Y_1), \ldots, (\mathcal{G}_N, Y_N)\}$ be a set of $N$ graphs with their real value properties or categories, where $\mathcal{G}_n$ refers to the $n$-th graph and $Y_n$ refers to the corresponding properties or labels. We denote by $\mathcal{G}_n = (\mathbb{V}, \mathbb{E}, \boldsymbol{A}, \boldsymbol{X})$ the $n$-th graph of size $\boldsymbol{M}_n$ with node set $\mathbb{V} = \{V_i | i = 1, \ldots, M_n\}$, edge set $\mathbb{E} = \{(V_i, V_j) | i > j; V_i, V_j \text{ is connected}\}$, adjacent matrix $\boldsymbol{A} \in \{0, 1\}^{M_n \times M_n}$, and feature matrix $\boldsymbol{X} \in \mathbb{R}^{M_n \times d}$ of $\boldsymbol{V}$ with $d$ dimensions, respectively. Denote the neighborhood of $V_i$ as $\mathcal{N}(V_i) = \{V_j | (V_i, V_j) \in \mathbb{E}\}$. We use $\mathcal{G}_{sub}$ as a specific subgraph and $\overline{\mathcal{G}}_{sub}$ as the complementary structure of $\mathcal{G}_{sub}$ in $\mathcal{G}$. Let $f : \mathbb{G} \to \mathbb{R}/[0, 1, \cdots, n]$ be the mapping from graphs to the real value property or category, $\boldsymbol{Y}$, $\mathbb{G}$ is the domain of the input graphs. $I(\boldsymbol{X}, \boldsymbol{Y})$ refers to the Shannon mutual information of two random variables.

### 3.1 GRAPH CONVOLUTIONAL NETWORK

Graph convolutional network (GCN) is widely adopted to graph classification. Given a graph $\mathcal{G} = (\mathbb{V}, \mathbb{E})$ with node feature $\boldsymbol{X}$ and adjacent matrix $\boldsymbol{A}$, GCN outputs the node embeddings $\boldsymbol{X}'$ from the following process:

$$\boldsymbol{X}' = \text{GCN}(\boldsymbol{A}, \boldsymbol{X}; \boldsymbol{W}) = \text{ReLU}(\boldsymbol{D}^{-\frac{1}{2}} \hat{\boldsymbol{A}} \boldsymbol{D}^{-\frac{1}{2}} \boldsymbol{X} \boldsymbol{W}), \tag{1}$$

where $\boldsymbol{D}$ refers to the diagonal matrix with nodes' degrees and $\boldsymbol{W}$ refers to the model parameters.

One can simply sum up the node embeddings to get a fixed length graph embeddings (Xu et al., 2019). Recently, researchers attempt to exploit hierarchical structure of graphs, which leads to various graph pooling methods (Li et al., 2019; Gao & Ji, 2019; Lee et al., 2019; Diehl, 2019; Zhang et al., 2018; Ranjan et al., 2020; Ying et al., 2018). Li et al. (2019) enhances the graph pooling with self-attention mechanism to leverage the importance of different nodes contributing to the results. Finally, the graph embedding is obtained by multiplying the node embeddings with the normalized attention scores:

$$\boldsymbol{E} = \text{Att}(\boldsymbol{X}') = \text{softmax}(\Phi_2 \tanh(\Phi_1 \boldsymbol{X}'^T)) \boldsymbol{X}', \tag{2}$$

where $\Phi_1$ and $\Phi_2$ refers to the model parameters of self-attention.

### 3.2 OPTIMIZING INFORMATION BOTTLENECK OBJECTIVE

Given the input signal $X$ and the label $Y$, the objective of IB is maximized to find the the internal code $Z$: $\max_Z I(Z, Y) - \beta I(X, Z)$, where $\beta$ refers to a hyper-parameter trading off informativeness and compression. Optimizing this objective will lead to a compact but informative $Z$. Alemi et al. (2017) optimize a tractable lower bound of the IB objective:

$$\mathcal{L}_{VIB} = \frac{1}{N} \sum_{i=1}^{N} \int p(z|x_i) \log q_\phi(y_i|z) dz - \beta \text{KL}(p(z|x_i)|r(z)), \tag{3}$$

where $q_\phi(y_i|z)$ is the variational approximation to $p_\phi(y_i|z)$ and $r(z)$ is the prior distribution of $Z$. However, it is hard to estimate the mutual information in high dimensional space when the distribution forms are inaccessible, especially for irregular graph data.

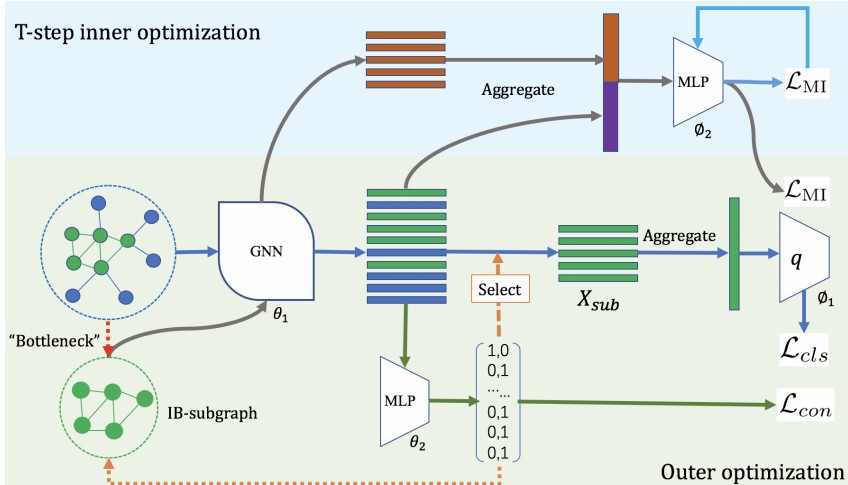

Figure 1: Illustration of the proposed graph information bottleneck (GIB) framework. We employ a bi-level optimization scheme to optimize the GIB objective and thus yielding the IB-subgraph. In the inner optimization phase, we estimate $I(\mathcal{G}, \mathcal{G}_{sub})$ by optimizing the statistics network of the DONSKER-VARADHAN representation (Donsker & Varadhan, 1983). Given a good estimation of $I(\mathcal{G}, \mathcal{G}_{sub})$, in the outer optimization phase, we maximize the GIB objective by optimizing the mutual information, the classification loss $\mathcal{L}_{cls}$ and connectivity loss $\mathcal{L}_{con}$.

# 4 OPTIMIZING THE GRAPH INFORMATION BOTTLENECK OBJECTIVE FOR SUBGRAPH RECOGNITION

In this section, we will elaborate the proposed method in details. We first formally define the graph information bottleneck and IB-subgraph. Then, we introduce a novel framework for GIB to effectively find the IB-subgraph. We further propose a bi-level optimization scheme and a graph mutual information estimator for GIB optimization. Moreover, we do a continuous relaxation to the generation of subgraph, and propose a novel loss to stabilize the training process.

## 4.1 GRAPH INFORMATION BOTTLENECK

We generalize the information bottleneck principle to learn an informative representation of irregular graphs, which leads to the graph information bottleneck (GIB) principle.

**Definition 4.1** (Graph Information Bottleneck). Given a graph $\mathcal{G}$ and its label $Y$, the GIB seeks for the most informative yet compressed representation $Z$ by optimizing the following objective:

$$\max_Z I(Y, Z) \text{ s.t. } I(\mathcal{G}, Z) \leq I_c. \tag{4}$$

where $I_c$ is the information constraint between $\mathcal{G}$ and $Z$. By introducing a Lagrange multiplier $\beta$ to Eq. 4, we reach its unconstrained form:

$$\max_Z I(Y, Z) - \beta I(\mathcal{G}, Z). \tag{5}$$

Eq. 5 gives a general formulation of GIB. Here, in subgraph recognition, we focus on a subgraph which is compressed with minimum information loss in terms of graph properties.

**Definition 4.2** (IB-subgraph). For a graph $\mathcal{G}$, its maximally informative yet compressed subgraph, namely IB-subgraph can be obtained by optimizing the following objective:

$$\max_{\mathcal{G}_{sub} \in \mathbb{G}_{sub}} I(Y, \mathcal{G}_{sub}) - \beta I(\mathcal{G}, \mathcal{G}_{sub}). \tag{6}$$

where $\mathbb{G}_{sub}$ indicates the set of all subgraphs of $\mathcal{G}$.

IB-subgraph enjoys various pleasant properties and can be applied to multiple graph learning tasks such as improvement of graph classification, graph interpretation, and graph denoising. However, the GIB objective in Eq. 6 is notoriously hard to optimize due to the intractability of mutual information and the discrete nature of irregular graph data. We then introduce approaches on how to optimize such objective and derive the IB-subgraph.

## 4.2 BI-LEVEL OPTIMIZATION FOR THE GIB OBJECTIVE

The GIB objective in Eq. 6 consists of two parts. We examine the first term $I(Y, \mathcal{G}_{sub})$ in Eq. 6, first. This term measures the relevance between $\mathcal{G}_{sub}$ and $Y$. We expand $I(Y, \mathcal{G}_{sub})$ as:

$$I(Y, \mathcal{G}_{sub}) = \int p(y, \mathcal{G}_{sub}) \log p(y|\mathcal{G}_{sub}) dy \, d\mathcal{G}_{sub} + \mathrm{H}(Y). \tag{7}$$

$H(Y)$ is the entropy of $Y$ and thus can be ignored. In practice, we approximate $p(y, \mathcal{G}_{sub})$ with an empirical distribution $p(y, \mathcal{G}_{sub}) \approx \frac{1}{N} \sum_{i=1}^{N} \delta_y(y_i) \delta_{\mathcal{G}_{sub}}(\mathcal{G}_{sub_i})$, where $\delta()$ is the Dirac function to sample training data. $\mathcal{G}_{sub_i}$ and $y_i$ are the output subgraph and graph label corresponding to i-th training data. By substituting the true posterior $p(y|\mathcal{G}_{sub})$ with a variational approximation $q_{\phi_1}(y|\mathcal{G}_{sub})$, we obtain a tractable lower bound of the first term in Eq. 6:

$$I(Y, \mathcal{G}_{sub}) \geq \int p(y, \mathcal{G}_{sub}) \log q_{\phi_1}(y|\mathcal{G}_{sub}) dy \, d\mathcal{G}_{sub}$$
$$\approx \frac{1}{N} \sum_{i=1}^{N} \log q_{\phi_1}(y_i|\mathcal{G}_{sub_i}) =: -\mathcal{L}_{cls}(q_{\phi_1}(y|\mathcal{G}_{sub}), y_{gt}), \tag{8}$$

where $y_{gt}$ is the ground truth label of the graph. Eq. 8 indicates that maximizing $I(Y, \mathcal{G}_{sub})$ is achieved by the minimization of the classification loss between $Y$ and $\mathcal{G}_{sub}$ as $\mathcal{L}_{cls}$. Intuitively, minimizing $\mathcal{L}_{cls}$ encourages the subgraph to be predictive of the graph label. In practice, we choose the cross entropy loss for categorical $Y$ and the mean squared loss for continuous $Y$, respectively. For more details of deriving Eq. 7 and Eq. 8, please refer to Appendix A.1.

Then, we consider the minimization of $I(\mathcal{G}, \mathcal{G}_{sub})$ which is the second term of Eq. 6. Remind that Alemi et al. (2017) introduces a tractable prior distribution $r(Z)$ in Eq. 3, and thus results in a variational upper bound. However, this setting is troublesome as it is hard to find a reasonable prior distribution for $p(\mathcal{G}_{sub})$, which is the distribution of graph substructures instead of latent representation. Thus we go for another route. Directly applying the DONSKER-VARADHAN representation (Donsker & Varadhan, 1983) of the KL-divergence, we have:

$$I(\mathcal{G}, \mathcal{G}_{sub}) = \sup_{f_{\phi_2}:\mathbb{G}\times\mathbb{G}\to\mathbb{R}} \mathbb{E}_{\mathcal{G},\mathcal{G}_{sub}\in p(\mathcal{G},\mathcal{G}_{sub})} f_{\phi_2}(\mathcal{G}, \mathcal{G}_{sub}) - \log \mathbb{E}_{\mathcal{G}\in p(\mathcal{G}),\mathcal{G}_{sub}\in p(\mathcal{G}_{sub})} e^{f_{\phi_2}(\mathcal{G},\mathcal{G}_{sub})}, \tag{9}$$

where $f_{\phi_2}$ is the statistics network that maps from the graph set to the set of real numbers. In order to approximate $I(\mathcal{G}, \mathcal{G}_{sub})$ using Eq. 9, we design a statistics network based on modern GNN architectures as shown by Figure 1: first we use a GNN to extract embeddings from both $\mathcal{G}$ and $\mathcal{G}_{sub}$ (parameter shared with the subgraph generator, which will be elaborated in Section 4.3), then concatenate $\mathcal{G}$ and $\mathcal{G}_{sub}$ embeddings and feed them into a MLP, which finally produces the real number. In conjunction with the sampling method to approximate $p(\mathcal{G}, \mathcal{G}_{sub})$, $p(\mathcal{G})$ and $p(\mathcal{G}_{sub})$, we reach the following optimization problem to approximate[1] $I(\mathcal{G}, \mathcal{G}_{sub})$:

$$\max_{\phi_2} \quad \mathcal{L}_{\mathrm{MI}}(\phi_2, \mathcal{G}_{sub}) = \frac{1}{N} \sum_{i=1}^{N} f_{\phi_2}(\mathcal{G}_i, \mathcal{G}_{sub_i}) - \log \frac{1}{N} \sum_{i=1,j\neq i}^{N} e^{f_{\phi_2}(\mathcal{G}_i, \mathcal{G}_{sub_j})}. \tag{10}$$

With the approximation to the MI in graph data, we combine Eq. 6, Eq. 8 and Eq. 10 and formulate the optimization process of GIB as a tractable bi-level optimization problem:

$$\min_{\mathcal{G}_{sub}, \phi_1} \quad \mathcal{L}(\mathcal{G}_{sub}, \phi_1, \phi_2^*) = \mathcal{L}_{cls}(q_{\phi_1}(y|\mathcal{G}_{sub}), y_{gt}) + \beta \mathcal{L}_{\mathrm{MI}}(\phi_2^*, \mathcal{G}_{sub}) \tag{11}$$
$$\text{s.t.} \quad \phi_2^* = \arg\max_{\phi_2} \mathcal{L}_{\mathrm{MI}}(\phi_2, \mathcal{G}_{sub}). \tag{12}$$

We first derive a sub-optimal $\phi_2$ notated as $\phi_2^*$ by optimizing Eq. 12 for T steps as inner loops. After the T-step optimization of the inner-loop ends, Eq. 10 is a proxy for MI minimization for the GIB objective as an outer loop. Then, the parameter $\phi_1$ and the subgraph $\mathcal{G}_{sub}$ are optimized to yield IB-subgraph. However, in the outer loop, the discrete nature of $\mathcal{G}$ and $\mathcal{G}_{sub}$ hinders applying the gradient-based method to optimize the bi-level objective and find the IB-subgraph.

---

[1]Notice that the MINE estimator (Belghazi et al., 2018) straightforwardly uses the DONSKER-VARADHAN representation to derive an MI estimator between the regular input data and its vectorized representation/encoding. It cannot be applied to estimate the mutual information between $\mathcal{G}$ and $\mathcal{G}_{sub}$ since both of $\mathcal{G}$ and $\mathcal{G}_{sub}$ are irregular graph data.

Table 1: Classification accuracy. The pooling methods yield pooling aggregation while the backbones yield mean aggregation. The proposed GIB method with backbones yields subgraph embedding by aggregating the nodes in subgraphs.

| Method | MUTAG | PROTEINS | IMDB-BINARY | DD |
|---|---|---|---|---|
| SortPool | **0.844 ± 0.141** | 0.747 ± 0.044 | 0.712 ± 0.047 | 0.732 ± 0.087 |
| ASAPool | 0.743 ± 0.077 | 0.721 ± 0.043 | 0.715 ± 0.044 | 0.717 ± 0.037 |
| DiffPool | 0.839 ± 0.097 | 0.727 ± 0.046 | 0.709 ± 0.053 | 0.778 ± 0.030 |
| EdgePool | 0.759 ± 0.077 | 0.723 ± 0.044 | 0.728 ± 0.044 | 0.736 ± 0.040 |
| AttPool | 0.721 ± 0.086 | 0.728 ± 0.041 | 0.722 ± 0.047 | 0.711 ± 0.055 |
| GCN | 0.743±0.110 | 0.719±0.041 | 0.707 ± 0.037 | 0.725 ± 0.046 |
| GraphSAGE | 0.743±0.077 | 0.721 ± 0.042 | 0.709 ± 0.041 | 0.729 ± 0.041 |
| GIN | 0.825±0.068 | 0.707 ± 0.056 | 0.732 ± 0.048 | 0.730 ± 0.033 |
| GAT | 0.738 ± 0.074 | 0.714 ± 0.040 | 0.713 ± 0.042 | 0.695 ± 0.045 |
| GAT + DropEdge | 0.743±0.081 | 0.711±0.043 | 0.710±0.041 | 0.717±0.035 |
| **GCN+GIB** | 0.776 ± 0.075 | 0.748 ± 0.046 | 0.722 ± 0.039 | 0.765 ± 0.050 |
| **GraphSAGE+GIB** | 0.760 ± 0.074 | 0.734 ± 0.043 | 0.719 ± 0.052 | **0.781 ± 0.042** |
| **GIN+GIB** | 0.839 ± 0.064 | **0.749 ± 0.051** | **0.737 ± 0.070** | 0.747 ± 0.039 |
| **GAT+GIB** | 0.749 ± 0.097 | 0.737 ± 0.044 | 0.729 ± 0.046 | 0.769 ± 0.040 |
| **GAT+GIB+DropEdge** | 0.754±0.085 | 0.737±0.037 | 0.731±0.003 | 0.776±0.034 |

Table 2: The mean and standard deviation of absolute property bias between the graphs and the corresponding subgraphs.

| Method | QED | DRD2 | HLM-CLint | MLM-CLint |
|---|---|---|---|---|
| GCN+Att05 | 0.48± 0.07 | 0.20± 0.13 | 0.90± 0.89 | 0.92± 0.61 |
| GCN+Att07 | 0.41± 0.07 | 0.16± 0.11 | 1.18± 0.60 | 1.69± 0.88 |
| **GCN+GIB** | **0.38± 0.12** | **0.06± 0.09** | **0.37± 0.30** | **0.72± 0.55** |

## 4.3 THE SUBGRAPH GENERATOR AND CONNECTIVITY LOSS

To alleviate the discreteness in Eq. 11, we propose the continuous relaxation to the subgraph recognition and propose a loss to stabilize the training process.

**Subgraph generator:** For the input graph $\mathcal{G}$, we generate its IB-subgraph with the node assignment $\boldsymbol{S}$ which indicates the node is in $\mathcal{G}_{sub}$ or $\overline{\mathcal{G}}_{sub}$. Then, we introduce a continuous relaxation to the node assignment with the probability of nodes belonging to the $\mathcal{G}_{sub}$ or $\overline{\mathcal{G}}_{sub}$. For example, the $i$-th row of $\boldsymbol{S}$ is a 2-dimensional vector $[p(V_i \in \mathcal{G}_{sub}|V_i), p(V_i \in \overline{\mathcal{G}}_{sub}|V_i)]$. We first use an $l$-layer GNN to obtain the node embedding and employ a multi-layer perceptron (MLP) to output $\boldsymbol{S}$ :

$$\boldsymbol{X}^l = \text{GNN}(\boldsymbol{A}, \boldsymbol{X}^{l-1}; \theta_1), \quad \boldsymbol{S} = Softmax(\text{MLP}(\boldsymbol{X}^l; \theta_2)). \tag{13}$$

$\boldsymbol{S}$ is a $n \times 2$ matrix, where $n$ is the number of nodes. We add row-wise Softmax to the output of MLP to ensure the nodes are either in or out of the subgraph. For simplicity, we compile the above modules as the subgraph generator, denoted as $g(; \theta)$ with $\theta := (\theta_1, \theta_2)$. When $\boldsymbol{S}$ is well-learned, the assignment of nodes is supposed to saturate to 0/1. The representation of $\mathcal{G}_{sub}$, which is employed for predicting the graph label, can be obtained by taking the first row of $\boldsymbol{S}^T \boldsymbol{X}^l$.

**Connectivity loss:** However, poor initialization will cause $p(V_i \in \mathcal{G}_{sub}|V_i)$ and $p(V_i \in \overline{\mathcal{G}}_{sub}|V_i)$ to be close. This will either lead the model to assign all nodes to $\mathcal{G}_{sub}$ / $\overline{\mathcal{G}}_{sub}$, or result that the representations of $\mathcal{G}_{sub}$ contain much information from the redundant nodes. These two scenarios will cause the training process to be unstable. On the other hand, we suppose our model to have an inductive bias to better leverage the topological information while $\boldsymbol{S}$ outputs the subgraph at a node-level. Therefore, we propose the following connectivity loss:

$$\mathcal{L}_{con} = ||\text{Norm}(\boldsymbol{S}^T \boldsymbol{A} \boldsymbol{S}) - \boldsymbol{I}_2||_F, \tag{14}$$

where $\text{Norm}(\cdot)$ is the row-wise normalization, $||\cdot||_F$ is the Frobenius norm, and $\boldsymbol{I}_2$ is a $2 \times 2$ identity matrix. $\mathcal{L}_{con}$ not only leads to distinguishable node assignment, but also encourage the subgraph to be compact. Take $(\boldsymbol{S}^T \boldsymbol{A} \boldsymbol{S})_{1:}$ for example, denote $a_{11}, a_{12}$ the element 1,1 and the element 1,2 of

Table 3: Ablation study on $\mathcal{L}_{con}$ and $\mathcal{L}_{MI}$. Note that we try several initiations for GIB w/o $\mathcal{L}_{con}$ and $\mathcal{L}_{MI}$ to get the current results due to the instability of optimization process.

| Method | QED | DRD2 | HLM-CLint | MLM-CLint |
|---|---|---|---|---|
| GIB w/o $\mathcal{L}_{con}$ | 0.46± 0.07 | 0.15± 0.12 | 0.45± 0.37 | 1.58± 0.86 |
| GIB w/o $\mathcal{L}_{MI}$ | 0.43± 0.15 | 0.21± 0.13 | 0.48± 0.34 | 1.20± 0.97 |
| **GIB** | **0.38± 0.12** | **0.06± 0.09** | **0.37± 0.30** | **0.72± 0.55** |

$\boldsymbol{S}^T \boldsymbol{A} \boldsymbol{S}$,

$$a_{11} = \sum_{i,j} A_{ij} p(V_i \in \mathcal{G}_{sub}|V_i) p(V_j \in \mathcal{G}_{sub}|V_j), a_{12} = \sum_{i,j} A_{ij} p(V_i \in \mathcal{G}_{sub}|V_i) p(V_j \in \overline{\mathcal{G}}_{sub}|V_j). \tag{15}$$

Minimizing $\mathcal{L}_{con}$ results in $\frac{a_{11}}{a_{11}+a_{12}} \to 1$. This occurs if $V_i$ is in $\mathcal{G}_{sub}$, the elements of $\mathcal{N}(V_i)$ have a high probability in $\mathcal{G}_{sub}$. Minimizing $\mathcal{L}_{con}$ also encourages $\frac{a_{12}}{a_{11}+a_{12}} \to 0$. This encourages $p(V_i \in \mathcal{G}_{sub}|V_i) \to 0/1$ and less cuts between $\mathcal{G}_{sub}$ and $\overline{\mathcal{G}}_{sub}$. This also holds for $\overline{\mathcal{G}}_{sub}$ when analyzing $a_{21}$ and $a_{22}$.

In a word, $\mathcal{L}_{con}$ encourages distinctive $\boldsymbol{S}$ to stabilize the training process and a compact topology in the subgraph. Therefore, the overall loss is:

$$\min_{\theta, \phi_1} \quad \mathcal{L}(\theta, \phi_1, \phi_2^*) = \mathcal{L}_{cls}(q_{\phi_1}(g(\mathcal{G};\theta)), y_{gt}) + \alpha \mathcal{L}_{con}(g(\mathcal{G};\theta)) + \beta \mathcal{L}_{\mathrm{MI}}(\phi_2^*, \mathcal{G}_{sub})$$
$$\text{s.t.} \quad \phi_2^* = \arg\max_{\phi_2} \mathcal{L}_{\mathrm{MI}}(\phi_2, \mathcal{G}_{sub}). \tag{16}$$

We provide the pseudo code in the Appendix to better illustrate how to optimize the above objective.

## 5 EXPERIMENTS

In this section, we evaluate the proposed GIB method on three scenarios, including improvement of graph classification, graph interpretation and graph denoising.

### 5.1 BASELINES AND SETTINGS

**Improvement of graph classification:** For improvement of graph classification, GIB generates graph representation by aggregating the subgraph information. We plug GIB into various backbones including GCN (Kipf & Welling, 2017), GAT (Velickovic et al., 2017), GIN (Xu et al., 2019) and GraphSAGE (Hamilton et al., 2017). We compare the proposed method with the mean/sum aggregation (Kipf & Welling, 2017; Velickovic et al., 2017; Hamilton et al., 2017; Xu et al., 2019) and pooling aggregation (Zhang et al., 2018; Ranjan et al., 2020; Ying et al., 2018; Diehl, 2019) in terms of classification accuracy. Moreover, we apply DropEdge (Rong et al., 2020) to GAT, namely GAT+DropEdge, which randomly drop 30% edges in message-passing at node-level. Similarly, we apply GIB to GAT+DropEdge, resulting in GAT+GIB+DropEdge. For fare comparisions, all the backbones for different methods consist of the same 2-layer GNN with 16 hidden-size.

**Graph interpretation:** The goal of graph interpretation is to find the substructure which shares the most similar property to the molecule. If the substructure is disconnected, we evaluate its largest connected part. We compare GIB with the attention mechanism (Li et al., 2019). That is, we attentively aggregate the node information for graph prediction. The interpretable subgraph is generated by choosing the nodes with top 50% and 70% attention scores, namely Att05 and Att07. GIB outputs the interpretation with the IB-subgraph. Then, we evaluate the absolute property bias (the absolute value of the difference between the property of graph and subgraph) between the graph and its interpretation. Similarly, for fare comparisons, all the backbones for different methods consist of the same 2-layer GNN with 16 hidden-size.

**Graph denoising:** We translate the permuted graph into the line-graph and use GIB and attention to 1) infer the real structure of graph, 2) classify the permuted graph via the inferred structure. We further compare the performance of GCN and DiffPool on the permuted graphs.

Table 4: Quantitative results on graph denoising. We report the classification accuracy (Acc), number of real edges over total real edges (Recall) and number of real edges over total edges in subgraphs (Precision) on the test set

| Method | GCN | DiffPool | GCN+Att05 | GCN+Att07 | **GCN+GIB** |
|---|---|---|---|---|---|
| Recall | - | - | 0.226±0.047 | 0.324± 0.049 | **0.493± 0.035** |
| Precision | - | - | 0.638± 0.141 | 0.675± 0.104 | **0.692 ±0.061** |
| Acc | 0.617 | 0.658 | 0.649 | 0.667 | **0.684** |

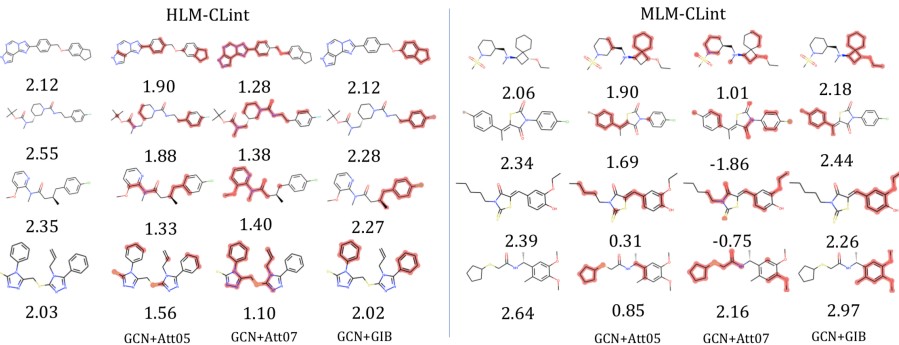

Figure 2: The molecules with their interpretable subgraphs discovered by different methods. These subgraphs exhibit similar chemical properties compared to the molecules on the left.

## 5.2 DATASETS

**Improvement of graph classification:** We evaluate different methods on the datasets of **MUTAG** (Rupp et al., 2012), **PROTEINS** (Borgwardt et al., 2005), **IMDB-BINARY** and **DD** (Rossi & Ahmed, 2015) datasets. [2]. The statistics of the datasets are available in Table 7 of Appendix.

**Graph interpretation:** We construct the datasets for graph interpretation on four molecule properties based on ZINC dataset, which contains 250K molecules. **QED** measures the drug likeness of a molecule, which is bounded within the range $(0, 1.0)$. **DRD2** measures the probability that a molecule is active against dopamine type 2 receptor, which is bounded with $(0, 1.0)$. **HLM-CLint** and **MLM-CLint** are estimated values of in vitro human and mouse liver microsome metabolic stability (base 10 logrithm of mL/min/g). We sample the molecules with QED $\geq 0.85$, DRD2 $\geq 0.50$, HLM-CLint $\geq 2$, MLM-CLint $\geq 2$ for each task. We use $85\%$ of these molecules for training, $5\%$ for validating and $10\%$ for testing. The statistics of the datasets are available in Table 8 of Appendix.

**Graph denoising:** We generate a synthetic dataset by adding $30\%$ redundant edges for each graph in **MUTAG** dataset. We use $70\%$ of these graphs for training, $5\%$ for validating and $25\%$ for testing.

## 5.3 RESULTS

**Improvement of Graph Classification:** In Table 1, we comprehensively evaluate the proposed method and baselines on improvement of graph classification. We train GIB on various backbones and aggregate the graph representations only from the subgraphs. We compare the performance of our framework with the mean/sum aggregation and pooling aggregation. This shows that GIB improves the graph classification by reducing the redundancies in the graph structure.

**Graph interpretation:** Table 2 shows the quantitative performance of different methods on the graph interpretation task. GIB is able to generate precise graph interpretation (IB-subgraph), as the substructures found by GIB has the most similar property to the input molecules. In Fig. 2, GIB generates more compact and reasonable interpretation to the property of molecules confirmed by chemical experts. More results are

Table 5: Average number of disconnected substructures per graph selected by different methods

| Method | QED | DRD2 | HLM | MLM |
|---|---|---|---|---|
| GCN+Att05 | 3.38 | 1.94 | 3.11 | 5.16 |
| GCN+Att07 | 2.04 | 1.76 | 2.75 | 3.00 |
| **GCN+GIB** | **1.57** | **1.08** | **2.29** | **2.06** |

[2]We follow the protocol in https://github.com/rusty1s/pytorch_geometric/tree/master/benchmark/kernel

Table 6: The influence of the hyper-parameter $\alpha$ of $L_{con}$ to the size of subgraphs.

| $\alpha$ | 1 | 3 | 5 | 10 |
|---|---|---|---|---|
| All | $0.483\pm0.143$ | $0.496\pm0.150$ | $0.494\pm0.147$ | $0.466\pm0.150$ |
| Max | $0.387\pm0.173$ | $0.413\pm0.169$ | $0.411\pm0.169$ | $0.391\pm0.172$ |

provided in the Appendix. In Table 5, we compare the average number of disconnected substructures per graph since a compact subgraph should preserve more topological information. GIB generates more compact subgraphs to better interpret the graph property. Moreover, compared to the baselines, GIB does not require a hyper-parameter to control the sizes of subgraphs, thus being more adaptive to different tasks. Please refer to Table 9 and Table 10 of Appendix for details. The training dynamic is shown in Fig. 7. We provide results with other MI estimators in Table 11 in Appendix.

**Graph denoising:** Table 4 shows the performance of different methods on noisy graph classification. GIB outperforms the baselines on classification accuracy by a large margin due to the superior property of IB-subgraph. Moreover, GIB is able to better reveal the real structure of permuted graphs in terms of precision and recall rate of true edges.

### 5.4 ABLATION STUDY

To further understand the rolls of $\mathcal{L}_{con}$ and $\mathcal{L}_{MI}$, we derive two variants of our method by deleting $\mathcal{L}_{con}$ and $\mathcal{L}_{MI}$, namely GIB w/o $\mathcal{L}_{con}$ and GIB w/o $\mathcal{L}_{MI}$. Note that GIB w/o $\mathcal{L}_{MI}$ is similar to InfoGraph (Sun et al., 2019) and GNNExplainer (Ying et al., 2019), as they only consider to maximize MI between latent embedding and global summarization and ignore compression. When adapted to sub graph recognition, it is likely to be $G = G_{sub}$. We evaluate the variants with 2-layer GCN and 16 hidden size on graph interpretation. In practice, we find that the training process of GIB w/o $\mathcal{L}_{con}$ is unstable as discussed in Section 4.3. Moreover, we find that GIB w/o $\mathcal{L}_{MI}$ is very likely to output $\mathcal{G}_{sub} = \mathcal{G}$, as it does not consider compression. Therefore, we try several initiations for GIB w/o $\mathcal{L}_{con}$ and $\mathcal{L}_{MI}$ to get the current results. As shown in Table 3, GIB also outperforms the variants, and thus indicates that every part of our model does contribute to the improvement of performance.

### 5.5 MORE DISCUSSION ON CONNECTIVITY LOSS

$L_{con}$ is proposed for stabilizing the training process and resulting in compact subgraphs. As it poses regularization for the subgraph generation, we are interested in its potential influence on the sizes of the chosen IB-subgraph. Therefore, we show the influence of different hyper-parameters of $L_{con}$ to the sizes of the chosen IB-subgraph. We implement the experiments with $\alpha$ varies in $\{1, 3, 5, 10\}$ on QED dataset and compute the mean and standard deviation of the sizes of IB-subgraph (All) and their largest connected parts (Max). As shown in Table 6, we observe that different $\alpha$ result in similar sizes of IB-subgraph. Therefore, its influence on the size of chosen subgraphs is weak.

## 6 CONCLUSION

In this paper, we have studied a subgraph recognition problem to infer a maximally informative yet compressed subgraph. We define such a subgraph as IB-subgraph and propose the graph information bottleneck (GIB) framework for effectively discovering an IB-subgraph. We derive the GIB objective from a mutual information estimator for irregular graph data, which is optimized by a bi-level learning scheme. A connectivity loss is further used to stabilize the learning process. We evaluate our GIB framework in the improvement of graph classification, graph interpretation and graph denoising. Experimental results verify the superior properties of IB-subgraphs.

### ACKNOWLEDGEMENTS

This work is partially funded by Beijing Natural Science Foundation (Grant No. JQ18017) and Youth Innovation Promotion Association CAS (Grant No. Y201929).

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

# A  APPENDIX

## A.1  MORE DETAILS ABOUT EQ. 7 AND EQ. 8

Here we provide more details about how to yield Eq. 7 and Eq. 8.

$$
\begin{aligned}
I(Y, \mathcal{G}_{sub}) &= \int p(y, \mathcal{G}_{sub}) \log p(y|\mathcal{G}_{sub}) dy \, d\mathcal{G}_{sub} - \int p(y, \mathcal{G}_{sub}) \log p(y) dy \, d\mathcal{G}_{sub} \\
&= \int p(y, \mathcal{G}_{sub}) \log p(y|\mathcal{G}_{sub}) dy \, d\mathcal{G}_{sub} + \mathrm{H}(Y) \\
&\geq \int p(y, \mathcal{G}_{sub}) \log q_{\phi_1}(y|\mathcal{G}_{sub}) dy \, d\mathcal{G}_{sub} + \mathrm{KL}(p(y|\mathcal{G}_{sub})|q_{\phi_1}(y|\mathcal{G}_{sub})) \\
&\geq \int p(y, \mathcal{G}_{sub}) \log q_{\phi_1}(y|\mathcal{G}_{sub}) dy \, d\mathcal{G}_{sub} \\
&\approx \frac{1}{N} \sum_{i=1}^{N} q_{\phi_1}(y_i|\mathcal{G}_{sub_i}) \\
&= -\mathcal{L}_{cls}(q_{\phi_1}(y|\mathcal{G}_{sub}), y_{gt})
\end{aligned}
\tag{17}
$$

## A.2  CASE STUDY

To understand the bi-level objective to MI minimization in Eq. 11, we provide a case study in which we optimize the parameters of distribution to reduce the mutual information between two variables. Consider $p(x) = sign(\mathcal{N}(0, 1)), p(y|x) = \mathcal{N}(y; x, \sigma^2)^3$. The distribution of $Y$ is:

$$
\begin{aligned}
p(y) &= \int p(y|x)p(x)dx \\
&= \sum_i p(y|x_i)p(x_i) \\
&= p(y|x = 1)p(x = 1) + p(y|x = -1)p(x = -1) \\
&= 0.5(\mathcal{N}(y; 1, \sigma^2) + \mathcal{N}(y; -1, \sigma^2))
\end{aligned}
\tag{18}
$$

We optimize the parameter $\sigma^2$ to reduce the mutual information between $X$ and $Y$. For each epoch, we sample 20000 data points from each distribution, denoted as $X = \{x_1, x_2, \cdots, x_{20000}\}, Y = \{y_1, y_2, \cdots, y_{20000}\}$. The inner-step is set to be 150. After the inner optimization ends, the model yields a good mutual information approximator and optimize $\sigma^2$ to reduce the mutual information by minimizing $L_{MI}$. We compute the mutual information with the traditional method and compare it with $L_{MI}$:

$$
\begin{aligned}
I(X, Y) &= \int p(x, y) \log \frac{p(y|x)}{p(y)} dx dy \\
&\approx \frac{1}{20000} \sum_{i=1}^{20000} \log \frac{p(y_i|x_i)}{p(y_i)}
\end{aligned}
\tag{19}
$$

As is shown in Fig .9, the mutual information decreases as $\mathcal{L}_{MI}$ descends. The advantage of such bi-level objective to MI minimization in Eq. 11 is that it only requires samples instead of forms of distribution. Moreover, it needs no tractable prior distribution for variational approximation. The drawback is that it needs additional computation in the inner loop.

## A.3  ALGORITHM

The algorithm is shown as following:

---

[3]We use the toy dataset from https://github.com/mzgubic/MINE

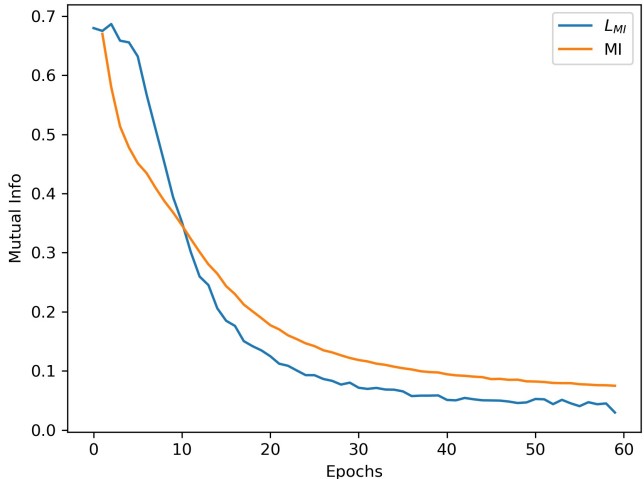

Figure 3: We use the bi-level objective to minimize the mutual information of two distributions. The MI is consistent with the loss as $\mathcal{L}_{MI}$ declines.

---

**Algorithm 1** Optimizing the graph information bottleneck.

---

**Input:** Graph $\mathcal{G} = \{A, X\}$, graph label $Y$, inner-step $T$, outer-step $N$.
**Output:** Subgraph $\mathcal{G}_{sub}$
1: **function** GIB($\mathcal{G} = \{A, X\}, Y, T, N$)
2:     $\theta \leftarrow \theta^0, \quad \phi_1 \leftarrow \phi_1^0$
3:     **for** $i = 0 \to N$ **do**
4:         $\phi_2 \leftarrow \phi_2^0$
5:         **for** $t = 0 \to T$ **do**
6:             $\phi_2^{t+1} \leftarrow \phi_2^t + \eta_1 \nabla_{\phi_2^t} \mathcal{L}_{\mathrm{MI}}$
7:         **end for**
8:         $\theta^{i+1} \leftarrow \theta^i - \eta_2 \nabla_{\theta^i} \mathcal{L}(\theta^i, \phi_1^i, \phi_2^T)$
9:         $\phi_1^{i+1} \leftarrow \phi_1^i - \eta_2 \nabla_{\phi_1^i} \mathcal{L}(\theta^i, \phi_1^i, \phi_2^T)$
10:    **end for**
11:    $\mathcal{G}_{sub} \leftarrow g(\mathcal{G}; \theta^N)$
12:    **return** $\mathcal{G}_{sub}$
13: **end function**

---

### A.4 MORE RESULTS ON GRAPH INTERPRETATION

In Fig. 4, we show the distribution of absolute bias between the property of graphs and subgraphs. GIB is able to generate such subgraphs with more similar properties to the original graphs.

In Fig. 5, we provide more results of four properties on graph interpretation.

### A.5 MORE RESULTS ON NOISY GRAPH CLASSIFICATION

We provide qualitative results on noisy graph classification in Fig. 6.

### A.6 DETAILS OF DATASETS

We provide the statistics of datasets in experiments. For graph classification, we evaluate the proposed method on four datasets, including MUTAG, PROTEINS, IMDB-BINARY and DD. The

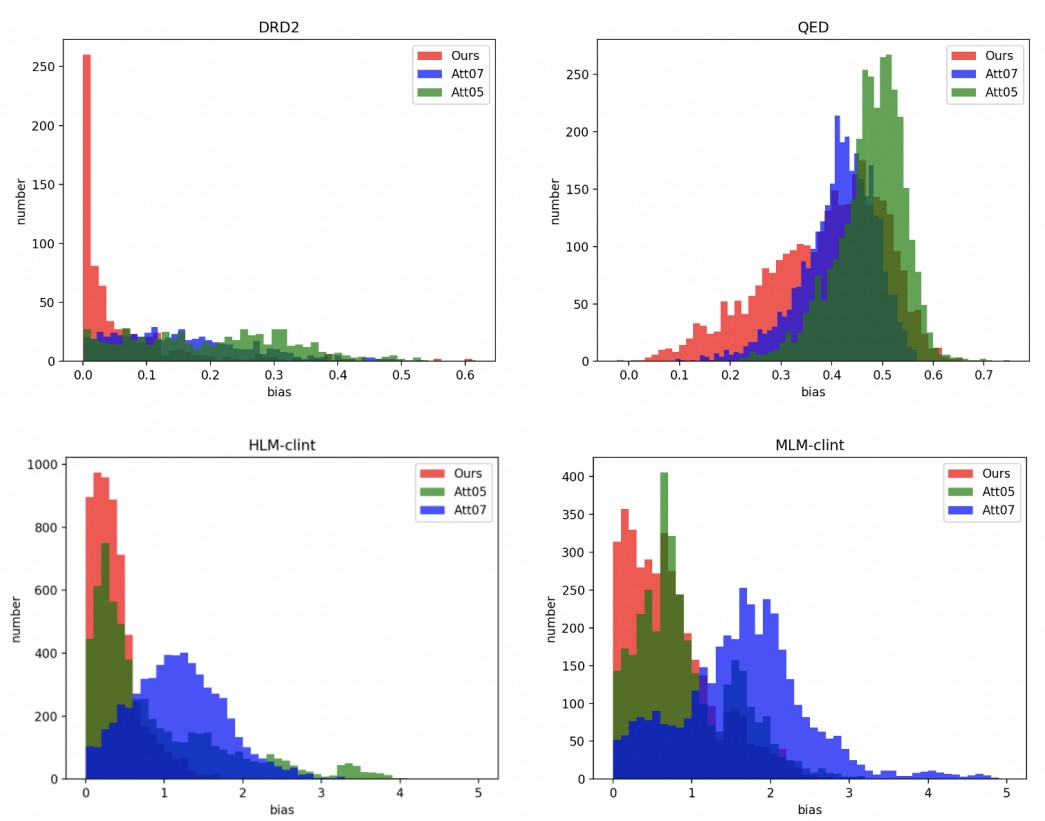

Figure 4: The histgram of absolute bias between the property of graphs and subgraphs.

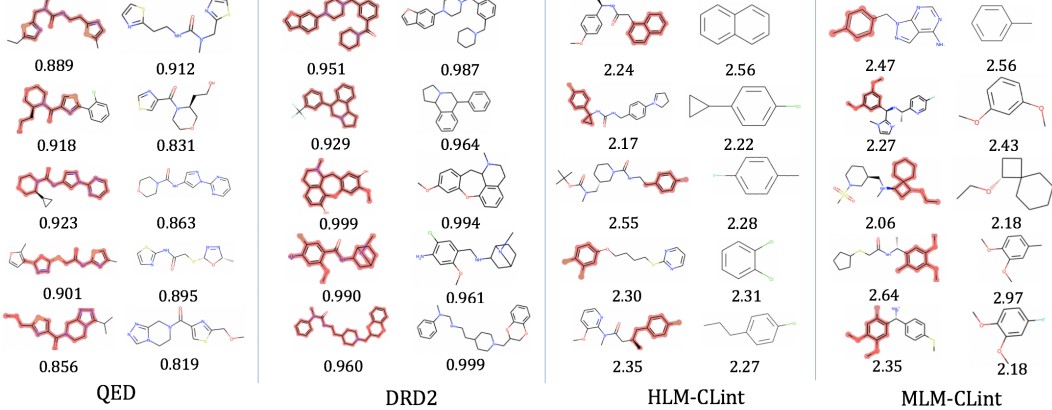

Figure 5: The molecules with its interpretation found by GIB. These subgraphs exhibit similar chemical properties compared to the molecules on the left.

statistics of these datasets are shown in Table 7 [4]. For graph interpretation, we preprocess the ZINC dataset and obtain four datasets, namely QED, DRD2, HLM-CLint and MLM-CLint. The details of these datasets are shown in Table 8. The synthetic dataset for graph denoising is basically generated from MUTAG dataset, please refer to Section 5.2 for details.

---

[4]The statistics of datasets in graph classification are collected from http://networkrepository.com

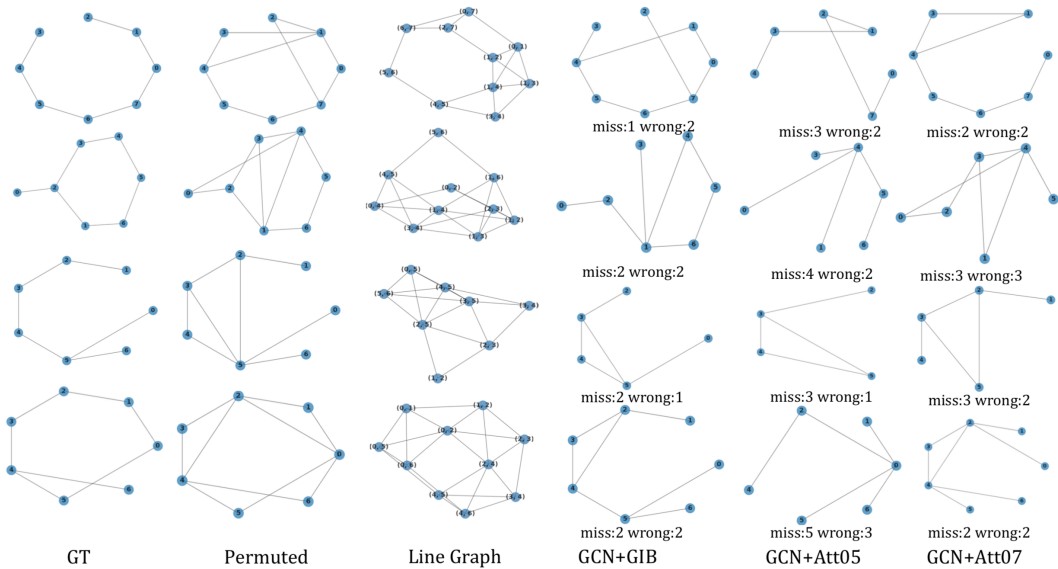

Figure 6: We show the blindly denoising results on permuted graphs. Each method operates on the line-graphs and tries to recover the true topology by removing the redundant edges. Columns 4,5,6 shows results obtained by different methods, where "miss: $m$, wrong: $n$" means missing $m$ edges and there are $n$ wrong edges in the output graph. GIB always recognizes more similar structure to the ground truth (not provided in the training process) than other methods.

Table 7: Statistics of datasets in improvement of graph classification.

|  | MUTAG | PROTEINS | IMDB-BINARY | DD |
|---|---|---|---|---|
| Nodes | 97.9K | 43.5K | 19.8K | 334.9K |
| Edges | 202.5K | 162.1K | 386.1K | 1.7M |
| Density | $4.2 \times 10^{-5}$ | $1.7 \times 10^{-4}$ | $2.0 \times 10^{-3}$ | $3.0 \times 10^{-5}$ |
| Maximum degree | 20 | 50 | 540 | 38 |
| Minimum degree | 2 | 2 | 4 | 2 |
| Average degree | 4 | 7 | 39 | 10 |
| Number of triangles | 2.8K | 366K | 18.8M | 7.1M |
| Average number of triangles | 0 | 8 | 951 | 21 |
| Maximum number of triangles | 12 | 136 | 17.8K | 160 |
| Average clustering coefficient | 0.001965 | 0.316645 | 0.831934 | 0.413379 |
| Fraction of closed triangles | 0.003160 | 0.315106 | 0.803561 | 0.410832 |
| Maximum k-core | 5 | 9 | 117 | 15 |
| Lower bound of Maximum Clique | 6 | 5 | 18 | 4 |

### A.7 SIZES OF THE CHOSEN SUBGRAPHS IN GRAPH INTERPRETATION

In the graph interpretation task, the hyper-parameter of $L_{con}$, $\alpha$, is set to be 5 on four datasets. We show the mean and standard deviation of the sizes of subgraphs in percent in Table 9 and Table 10. Note that the sizes of chosen subgraphs mainly depend on task relevant information. For example, as DRD2 measures the probability of being active against dopamine type 2 receptor, it depends on almost the whole structure of a molecule. In contrast, HLM-CLint measures vitro human micro-some metabolic stability, which is greatly influenced by small motifs. As shown in Table 9 and Table 10, GCN+GIB can recognize the subgraphs with adaptive sizes on different tasks, leading to better performance. However, in GCN+Att05 and GCN+Att07, the size of subgraphs is explicitly controlled by the hyper-parameter (preserve top 50% or 70 % nodes with the highest attention scores). Therefore, the performances of these methods are limited.

Table 8: Statistics of datasets in graph interpretation.

|  | QED | DRD2 | HLM-CLint | MLM-CLint |
|---|---|---|---|---|
| Number of graphs | 35000 | 3000 | 25850 | 16666 |
| Maximum number of nodes | 29 | 66 | 37 | 37 |
| Minimum number of nodes | 12 | 13 | 9 | 7 |
| Average number of nodes | 21.82 | 27.43 | 25.14 | 22.44 |
| Maximum number of edges | 34 | 74 | 42 | 42 |
| Minimum number of edges | 12 | 14 | 9 | 7 |
| Average number of edges | 23.43 | 30.24 | 22.23 | 24.19 |
| Dimension of node features | 9 | 8 | 9 | 9 |

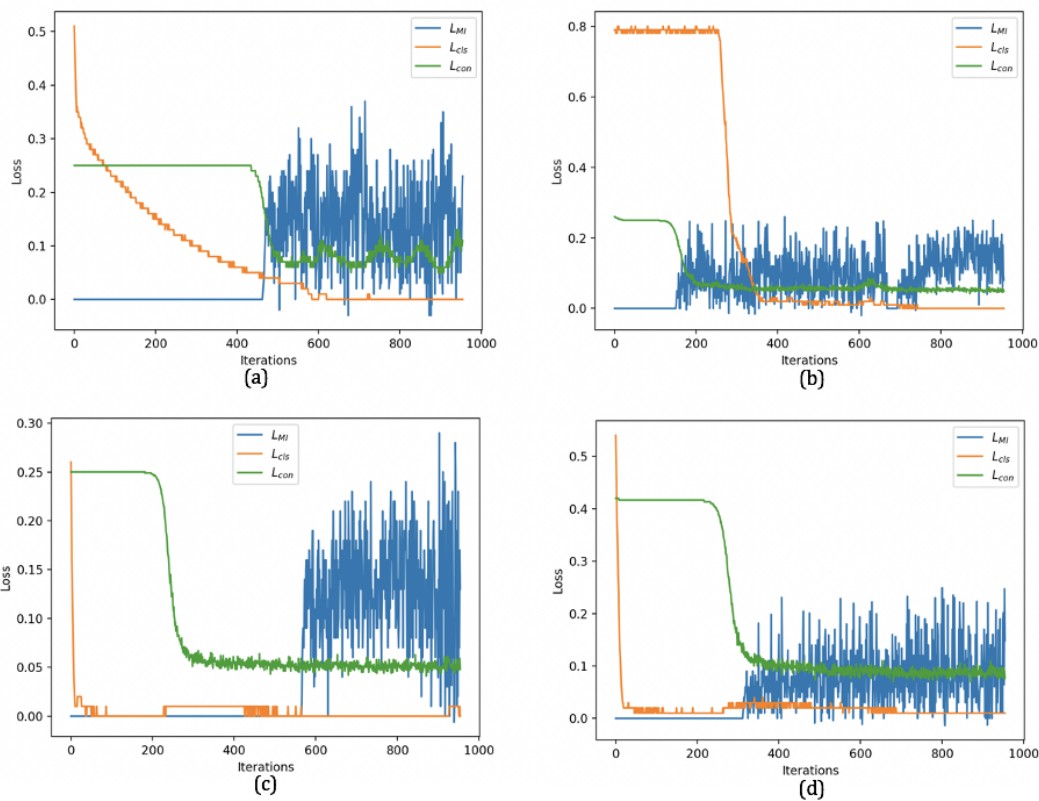

Figure 7: From (a) to (d), we show the loss in the training process on QED, DRD2, HLM-CLint and MLM-CLint respectively.

## A.8 THE TRAINING DYNAMIC

We show the loss in the training process in Figure. 7.

## A.9 IMPLEMENTATION WITH OTHER MUTUAL INFORMATION ESTIMATORS

As shown in (Sun et al., 2019; Nowozin et al., 2016), the f-divergence family can also approximate the mutual information. Here we provide the results of GCN+GIB with Jensen-Shannon Divergence (JSD) and $\chi^2$ Divergence ($\chi^2$) on graph classification in Table 11. Experiment results show that our model can also employ other mutual information estimators for bilevel optimization.

Table 9: Size of the chosen subgraphs on four datasets in percent.

| Method | QED | DRD2 | HLM-CLint | MLM-CLint |
|---|---|---|---|---|
| GCN+Att05 | 43.5±5.4 | 46.8±3.1 | 48.1±4.1 | 45.1±4.1 |
| GCN+Att07 | 65.8±3.4 | 66.7±3.0 | 65.8±5.7 | 67.6±5.5 |
| GCN+GIB | 49.6±15.0 | 94.8±5.3 | 47.7±13.7 | 54.7±17.2 |

Table 10: Size of largest connected parts used for graph interpretation in percent.

| Method | QED | DRD2 | HLM-CLint | MLM-CLint |
|---|---|---|---|---|
| GCN+Att05 | 22.5±9.5 | 34.7±9.4 | 23.5±7.2 | 29.7±7.9 |
| GCN+Att07 | 43.3±11.8 | 54.2±13.0 | 45.0±15.2 | 41.2±8.4 |
| GCN+GIB | 41.3±16.9 | 92.8±10.4 | 29.1±10.6 | 36.9±16.2 |

## A.10 INFLUENCE OF INITIALIZATION

As the initialization of our model may potentially influence the final chosen subgraphs, we rerun our model five times on the QED dataset for graph interpretation task. Then, we employ the intersection over union ($IoU$) to measure the overlap between the subgraphs in 5 different runs and the results reported in Table 2. Similarly, we compute the $IoU$ between the chosen subgraphs and their largest connected parts separately, which refer to $IoU_{all}$ and $IoU_{max}$. We finally report the mean and standard deviation of $IoU_{all}$, $IoU_{max}$ on the testing set in Table 12. We notice that different initialization has limited influence on the chosen subgraphs, as all the results of five additional runs have high portions of common nodes with the initial run.

Table 11: Classification accuracy on graph classification. We implement the mutual information estimator with Jensen-Shannon Divergence (JSD) and $\chi^2$ Divergence ($\chi^2$)

| Method | QED | DRD2 | HLM-CLint | MLM-CLint |
|---|---|---|---|---|
| GCN | 0.743± 0.110 | 0.719± 0.041 | 0.707± 0.037 | 0.725± 0.046 |
| GCN+GIB(DV) | 0.776± 0.075 | 0.748± 0.046 | 0.722± 0.039 | 0.765± 0.050 |
| GCN+GIB(JSD) | 0.758± 0.087 | 0.741± 0.040 | 0.718± 0.044 | 0.759± 0.057 |
| GCN+GIB($\chi^2$) | 0.756± 0.097 | 0.746± 0.057 | 0.721± 0.033 | 0.755± 0.049 |

Table 12: The overlap between the chosen subgraphs with different initialization.

| Run | 1 | 2 | 3 | 4 | 5 |
|---|---|---|---|---|---|
| $IoU_{all}$ | 0.848±0.163 | 0.765±0.106 | 0.784±0.112 | 0.829±0.166 | 0.813±0.186 |
| $IoU_{max}$ | 0.779±0.330 | 0.696±0.310 | 0.742±0.333 | 0.757±0.335 | 0.762±0.304 |

