# OpenReview forum: "Graph Information Bottleneck for Subgraph Recognition"
_ICLR.cc/2021/Conference — ICLR 2021 Poster_

### Official Review · AnonReviewer2 · 2020-10-27
**Interesting Information Theoretic Objective Functions for Graph Learning**

**Rating:** 7
**Confidence:** 3

**Review:**

Summary:

I think this is a nice paper that successfully used information theoretic objective functions for graph representation learning. The authors leveraged the DONSKER approximation of mutual information for a global information bottleneck loss used on the input-space instead of learned latent-space. To help stabilise optimisation, the authors also use bi-level optimisation with more iterations on the inner $I(G, G_{sub})$ as well as automatic masks learning through their $L_{con}$ loss. The authors also showed many experiments on graph classification, denoising, and interpretation tasks.

Reason for the Score:

Yet, I think the authors could improve on the related work coverage. For example, I found a paper published in last year's ICLR that seems quite related but not cited or compared in this paper. I hope the authors could spend a bit more time to add the relevant methods and possibly compare against them.

Pro:
- Quite interesting information theoretic objective functions that actually work on multiple graph learning tasks.
- Figure 1 is quite helpful for understanding the story quickly.
- Relatively well written and easy to follow the ideas.

Con / Questions:
- In Eq.13. Given that $\textbf{$\textit{S}$}$ are probability values, did the authors add a non-linearity such as sigmoid after the MLP ?
- The proposed method seems quite related to last year's ICLR paper $\textit{InfoGraph}$ which I think the authors should probably cite and maybe compare and contrast as well. For example, your proposed Eq.11 seems quite similar to $\textit{InfoGraph}$'s Eq.6. (https://openreview.net/pdf?id=r1lfF2NYvH)


-----------------------------------------------------------------
Post Rebuttal:

Many thanks for the authors to update their original paper addressing my questions and concerns.
I have now updated the score.

---

> ### Author Response · Authors · 2020-11-18
> **Reply**
>
> We thank the reviewers for valuable feedback and suggestions.
>
> Q1: Did the authors add a non-linearity such as sigmoid after the MLP?
> A1: Yes, we add row-wise Softmax to make $S$ contains reasonable probability values. Moreover, row-wise Softmax guarantees that $p(V_{i} \in G_{sub} |V_{i})+p(V_{i} \in \overline G_{sub} |V_{i}) = 1$. We have added these details in the updated version.
>
> Q2: Compare to InfoGraph.
> A2: Thank you for your suggestions. That is a good question and we will add this work in the related work. The intuition of InfoGraph is that a good global representation should correlate to multi-scale local representations [1] (e.g. nodes, edges and triangles). Hence, InfoGraph maximizes the mutual information (MI) of global and local representations. To do that, it uses Jensen-Shannon MI estimator [2] in InfoGraph’s Eq.5 to estimate MI. As they aim to maximize the MI, they directly optimize Eq.5 with the other loss, leading to Eq.6. Our work recognizes the IB-Subgraph by optimizing the objective in Eq.6. For the minimization of $I(G, G_{sub})$, we first estimate the MI with Donsker-Varadhan representation and minimize the MI by bilevel optimization. The same thing of these two MI estimator is that they both use an empirical distribution to approximate joint distribution and use permuted pairs to approximate samples from production of marginal distribution. The difference is that DV estimator is a stronger bound than that of JS estimator (note that JS estimator is one kind of f-divergence). Moreover, DV estimator is strong consistency. (Please refer to Eq.8, Lemma1, Lemma2 and Theorem 2 in MINE [3] for more details.)
>
> Can the MI estimator in InfoGraph be applied to our method? The answer is yes and we provide results with JSD and $\chi^{2}$ estimator in [2] on graph classification below.
>
> |Method | MUTAG | PROTEINS |IMDB-BINARY | DD |
> | :----------------  | :--------------- | : -------------  | :----------  | :--------------- |
> |GCN | 0.743+- 0.110 | 0.719 +- 0.041| 0.707 +- 0.037 | 0.725 +- 0.046|
> |GCN+GIB(DV) | 0.776 +- 0.075 |0.748 +- 0.046 | 0.722 +- 0.039 |0.765+-0.050 |
> |GCN+GIB(JSD) | 0.758 +- 0.087 |0.741+- 0.040 |0.718 +- 0.044| 0.759 +- 0.057|
> |GCN+GIB($\chi^{2}$)| 0.756+-0.097 |0.746+-0.057 | 0.721+-0.033 |0.755+-0.049 |
>
>
> [1] InfoGraph: Unsupervised and Semi-supervised Graph-Level Representation Learning via Mutual Information Maximization, Fan-Yun Sun et al, ICLR2020.
>
> [2] f-GAN: Training Generative Neural Samplers using Variational Divergence Minimization, Sebastian Nowozin et al, NeurIPS2016.
>
> [3] Mutual Information Neural Estimation, Mohamed Ishmael Belghazi et al, ICML2018.

---

### Official Review · AnonReviewer3 · 2020-10-28
**The idea is lack of novelty. The contribution is insufficient. A Clear Rejection.**

**Rating:** 3
**Confidence:** 4

**Review:**

The submission proposed to use Graph Information Bottleneck (GIB) for the subgraph recognition problem in deep graph learning. Basically, it makes use of bi-level optimization to find a subgraph that well encode the information for graph classification task. The major claim is that the resulting subgraph is more robust for the learning.

1). My major concern is on the novelty part. The submission is lack of novelty. First, the Graph Information Bottleneck (GIB) is used to learn a robust GNN against adverbial attack in the paper accepted in this year NeurIPS.
Graph Information Bottleneck
Tailin Wu · Hongyu Ren · Pan Li · Jure Leskovec
NeurIPS 2020.
The only difference is the submission uses the GIB principal to learn a subgraph. As for the subgraph selection, much work can be found. Basically, the edge dropping or node dropping  is popularly studied recently. By simple search, we can find DropEdge in ICLR’20, NeuralSparse in ICML’20, DropNode in NeurIPS’20. The only difference is they are modeling general GNN, while this submission is only focusing on graph classification. As for the subgraph selection, is there any superiority of the proposed methods over NeuralSparse which uses gumbel-softmax?

DropEdge: DropEdge: Towards Deep Graph Convolutional Networks on Node Classification
ICLR’20.

NeuralSparse: Robust Graph Representation Learning via Neural Sparsification.
ICML’20.

DropNode: A Flexible Generative Framework for Graph-based Semi-supervised Learning
NeurIPS’20.



2). The submission is not well presented. Many notations are not defined before use. For example, in 4.2,
δyi (y)δ(Gsub,i), what is meaning of δ? What is the meaning of G_sub, i ?


3). The experiments are not sufficient. The list methods are most famous GNNs but not SOTAs for graph classification task. Graph neural tangent kernel (GNTK), End-to-end graph classification (DCGNN) , Convolutional network for graphs (PATCHY-SAN). Besides, the graph kernel methods, like Graphlet kernel (GK), Weisfeiler-Lehman Graph Kernels (WLGK), and Propagation kernel (PK) are not compared.

DCGNN: Zhang, Muhan, Cui, Zhicheng, Neumann, Marion, & Chen, Yixin. 2018. An End-to-End Deep Learning Architecture for Graph Classification. In: The Thirty-Second AAAI Conference on Artificial Intelligence

4). The results reported seems much worse than results reported in other paper. For example, in the paper below. We can easily see the best result in MUTAG is 93.28±3.36, while the submission gives only 0.844 ± 0.141, in Proteins, best result is 77.47±4.34 while the submission gives only 0.749 ± 0.051. Similar case shows in DD dataset.

Structural Landmarking and Interaction Modelling: on Resolution Dilemmas in Graph Classification,
https://arxiv.org/pdf/2006.15763.pdf

5). Typos:
In 4.1, “a informative representation”==>”an informative representation”
In 4.3, “S is 2-dimensional vector”==>”S is a 2-dimensional vector”


6). The authors does not report or discuss the running time complexity of the algorithm. Since the framework needs bi-level optimization, it is supposed to discuss how fast the algorithm will converge.

---

> ### Author Response · Authors · 2020-11-18
> **Reply (Part 1 of 2)**
>
> We thank the reviewers for your valuable comments and suggestions.
>
> Q1: Lack of novelty and comparison with more work.
> A1: We restate that the contribution of our work is to recognize a meaningful subgraph from the originial graph in terms of graph labels or properties. This is a bright new research challenge in graph learning and the other works have not addressed it (we will discuss the differences of the proposed method to the other works in the next paragraph). It is like the object recognition to the computer vision. Subgraph recognition is interesting and important to graph learning because in many scenarios, we need to obtain a subgraph as informative as possible yet contains less redundancy and noise. Attention-based methods somehow are attentive to the important nodes but do not resolve the problem well. Therefore, We introduce the IB-Subgraph to resolve this problem. IB-Subgraph is able to facilitate many tasks, such as functional group discovery, graph classification, graph denoising and so on.  We also show that the existing GNNs with IB-Subgraph on graph classification will not break their expressive powers, but will improve the performances of themselves.
>
> As the reviewer lists many works in the feedback, we compare the proposed method with them and emphasize the novelty of our work.
> A1.1. Compare with GIB (Graph information bottleneck, Tailin Wu et al). We also noticed this parallel work in NeurIPS 2020 but we did not know the details of GIB until it was available on arXiv on October 24, 2020. Since GIB-SR (our work) was publicly available on Openreview.net on October 3, 2020, it is common sense that GIB and GIB-SR are parallel works. We carefully read the GIB paper and point out the main differences between GIB and GIB-SR (Our work) in the public comments.
>
> A1.2. Compare with DropEdge [1]. Two methods focus on two different problems. DropEdge reliefs over-smoothing of node embeddings in GNNs by randomly dropping a portion of edges in graphs to improve node classification. Meanwhile, our work juices out a compressed yet informative subgraph, namely IB-Subgraph, to facilitate graph-level tasks. But it is interesting to integrate DropEdge to our framework. Therefore, we added some experiments for GAT+GIB+DropEdge. As shown in the table, GAT+DropEdge with ad-hoc GIB-SR improves the performances by around 3%. Please refer to Table 1 in our updated version for more details.
>
> Table 1: More results on graph classification.
>
> |Method                       |     MUTAG       |   PROTEINS    |IMDB-BINARY |        DD            |
> | :----------------             | :---------------     | : -------------     | :----------             | :--------------- |
> |GAT                             | 0.738 +- 0.074 | 0.714 +- 0.040| 0.713 +- 0.042 | 0.695 +- 0.045|
> |GAT + DropEdge      |  0.743 +- 0.081 |0.711 +- 0.043 | 0.710 +- 0.041 |0.717+-0.035  |
> |GAT+GIB                    | 0.749 +- 0.097 |0.737+- 0.044   |0.729 +-  0.046| 0.769 +- 0.040|
> |GAT+GIB+DropEdge| 0.754+-0.085   |0.737+-0.037    | 0.731+-0.043  |0.776+-0.034   |
>
> A1.3. Compare with NeuralSparse [2]. It is also a method for node classification. On the other hand, NeuralSparse select Top-K (K is a hyper-parameter) task-relevant 1-hop neighbors of a central node, while we discover the IB-Subgraph without regularization to its scale (No Top-K), thanks to our information-theoretic objective. Therefore, NeuralSparse is more comparable to GIB (Tailin Wu et. al.). Moreover, our method discovers an informative substructure in a larger searching space (O(n^{K+1} at most) for NeuralSparse vs O(2^{n}) for our work, where n is the node number of a graph).
>
> A1.4. Compare with DropNode. It seems that ‘A flexible generative framework for graph based semi-supervised learning[3]’ does not use DropNode strategy. It studied semi-supervised learning, while we focus on discovering a compressed yet informative subgraph in a graph. GraphSAGE[4], FastGCN[5] and ASGCN[6] can be categorized as DropNode, because they randomly drop some neighborhoods in message passing[1]. Although they probably relief the over-smoothing phenomenon when GCN goes deeper, DropNode can not discover the informative subgraphs as it does not aim to identify which substructure is informative.
>
> A1.5. Why do not use Gumbel-Softmax in subgraph selection? Please refer to A1.3. We do not require a hyper-parameter (K) to restrict the size of IB-Subgraph. We mentioned similar issue when introducing the baseline GCN+Att05/07 in our paper.
>
> Q2: Notations.
> A2: The $\delta()$ function is known as the sampling function or Dirichlet function. It is used for representing the sampling procedure of training data due to its property. Similar to $y_{i}$, $G_{sub,i}$ means the i-th sample of $G_{sub}$ in a batch. We have clarified these issues to make our paper more self-contained.

---

> > ### Author Response · Authors · 2020-11-18
> > **Reply (Part 2 of 2)**
> >
> > Q3: Experiments are not sufficient.
> > A3.1: The major contribution of our work is to recognize meaningful subgraphs in terms of graph labels or properties.  We provide theoretical and empirical analysis on the validity of IB-Subgraph for various graph-level tasks.  On the other hand, our method is model-agnostic and is easy to plug into various GNNs, as long as it generates node-embeddings. By applying our method to various famous GNN backbones, we see significant improvement over these backbones. For a fair comparison, we implement each method with the same network architecture to its backbone (2 convolution layers with 16 hidden size).
> >
> > A3.2: We indeed compare with DCGNN in graph classification and cite this work. We refer it to SortPool, as it is well-known for the sortpool layer.
> >
> > Q4: The experiment results seem worse than a preprint work on graph classification.
> > A4: We emphasize that our work is not particularly proposed for graph classification. Instead, we focus on subgraph recognition. That is, to discover an informative yet compressed subgraph in graph data, namely IB-Subgraph. Such IB-Subgraph has many applications in graph learning as shown in our paper. And to improve the graph classification is one of its applications. We have shown that our method is model-agnostic and is easy to plug into various GNNs as long as it generates node-embeddings. We observe improvements of these baselines with the assistance of  IB-Subgraph. To be fair, our paper just implements each method with the same network architecture (2 convolution layers and hidden size of 16). Meanwhile, we must notice that the expressive power of different baselines also influence the performance. Hence it is hard to fairly compare the our method with the method in http://arxiv.org/pdf/2006.15763.pdf. However, the experiment results are sufficient to show the validity of our method, as we observe improvements of the baselines with the assistance of  IB-Subgraph on graph classification.
> >
> > Moreover, as the method in http://arxiv.org/pdf/2006.15763.pdf is particularly designed for graph classification. We find it incapable to obtain an informative yet compressed subgraph in graph data for graph interpretation and graph denoising tasks in our paper.
> >
> > Q5: Typo issues.
> > A5: We have corrected the typos in the revised version.
> >
> > Q6: Computational complexity.
> > A6: The additional computation complexity of our method compare to our backbones in every iteration is O(T), where T is the inner steps to update an MLP. Although our work takes more time in training compared to GNNs, it is proposed to recognize the IB-Subgraph from an information-theoretic perspective while GNNs are unable to recognize such substructure. IB-Subgraph enjoys good property and thus facilitates multiple graph-level tasks, including graph classification.
> >
> > [1]DropEdge: Towards Deep Graph Convolutional Networks on Node Classification, Yu Rong et al, ICLR2020.
> >
> > [2]NeuralSparse: Robust Graph Representation Learning via Neural Sparsification, Cheng Zheng et al, ICML2020.
> >
> > [3]A Flexible Generative Framework for Graph-based Semi-supervised Learning, Jiaqi Ma et al, NeurIPS2020.
> >
> > [4]Inductive representation learning on large graphs, Williams L. Hamilton et al, NeurIPS2017.
> >
> > [5]FastGCN: Fast learning with graph convolutional networks via importance sampling, Jie Chen et al, ICLR2018.
> >
> > [6]ASGCN: Adaptive sampling towards fast graph representation learning, Wenbing Huang et al, NeurIPS2018.

---

### Official Review · AnonReviewer4 · 2020-10-29
**A good paper with a clear theoretical contribution and rigorous empirical evaluation - clear accept recommendation with medium reviewer confidence**

**Rating:** 8
**Confidence:** 2

**Review:**

# Summary
The paper introduces the Graph Information Bottleneck (GIB) which aims to learn the most-informative compressed representation $Z$ given graph $G$ with associated label $Y$. Further, it defines GIB-Subgraph which aims to learn the compressed representation as the subgraph $G_{sub}$ which maximizes the mutual information within the family of subgraphs ${\cal G}_sub$ of $G$. The paper introduces bi-level optimization objective which has the following parts:

(a) optimizing the mutual information loss $L_{cls}$ between the subgraph representation $G_{sub}$ and the graph label $Y$ using the backbone GNN followed by aggregation of subgraph node embeddings $X_{sub}$ and cross-entropy loss when comparing to graph labels.

(b) approximates the mutual information $L_{MI}$ between the original graph and a subgraph $I(G, G_{sub})$ using statistics network $f_{\phi}$ which uses the backbone GNN to obtain graph embeddings (using mean/sum or pooling over node embeddings) followed by MLP over concatenated embeddings of $G$ and $G_{sub}$.

The procedure retrains the graph-subgraph mutual information estimator in the inner loop for each step (eqn. 10) before updating the parameters of the backbone GNN and the subgraph selection MLP and finally updating the subgraph-label MI estimator ($L_{cls}$). In order to obtain compact subgraphs, the paper introduces a regularization term $L_{con}$ closely related to graph cut.

The papers shows empirically on downstream task of graph classification that adding the GIP objective improves classification accuracy. Further, on graph interpretation task, the authors show that the GIP objective improves the similarity of the retrieved subgraphs using domain-specific metrics. The authors also evaluate on graph denoising on the MUTAG dataset.

# Recommendation
I vote for a strong accept. This paper is well-written, makes a clear theoretical contribution to the field as well as provides sufficient empirical evaluation.

# Questions to the authors
- I would have liked to see in the supplementary material an example of the algorithm on a toy graph example (similar to case study A).

- I wonder does the initialization have an influence on the final chosen subgraph nodes. Does $S$ (node-assignment) (always/almost always?) saturate  as mentioned on page 5?

- What is the influence of the ${\cal L}_{con}$ on the size of the final chosen subgraph. A table showing the size of final subgraphs (in term of output of MLP $\theta_2$ in Figure 1) might be helpful, though this is partially addressed in Table 4.

- For completeness, it would be good to provide in the supplementary material the properties of the datasets used e.g., number of graphs, mean/max/min number of nodes, edges, dimension of node features, dimension of edge features (if any), etc.

- It would have been good to see plots showing the convergence of the different losses as part of the bi-level optimization iterations.

- [optional] On the graph denoising experiment, it might be good to add more concrete evaluation both on larger graphs e.g. on graph families such as Power-Law, SBM as well as non-uniform edge addition.

---

> ### Author Response · Authors · 2020-11-18
> **Reply (Part 1 of 2)**
>
> We thank the reviewers for your valuable comments and suggestions.
>
> Q1: An example of the algorithm on a toy graph example (Similar to case study A).
> A1: This is a very good idea and we tried to implement it. However, when two random variables, X and Y, become graphs,  it is hard to construct graph datasets X, Y with some probability distribution assumptions and a correlation assumption between X and Y. Recall the case study A,  we have to compute the actual values of mutual information (MI) for each epoch in the optimization process to validate if the proposed method really reduces the true MI. Therefore, we design two random variables X and Y for the toy experiment instead so that it is possible to compute the real MI $I(X,Y)$ of X and Y.  Then, the toy example shows that the proposed bi-level method is able to reduce the real MI $I(X,Y)$.
>
> Q2: The influence of initialization on the final chosen subgraphs. And saturation of assignment S.
> A2: Initialization has little influence on the chosen of subgraph nodes. We added an experiment on QED dataset to see how the initialization affects the chosen of subgraphs. As shown in Table 1, the intersection over unions ($IoU$) between the subgraphs in the paper and the new subgraphs from 5 different random initializations indicate a consistent selections of the subgraph, where ''All'' denotes the whole chosen subgraph, and "Max" denotes the largest connected part of a subgraph used for property evaluation. The saturation is guaranteed by the antidiagonal of $(Norm(S^{T}AS)-I_{2})$. Note that the saturation here means distinctive assignment of nodes. The element of S does not definitely equal to 0/1, but still assign the node to subgraph or not with a high probability.
>
> Table 1: The overlap between the chosen subgraphs with different initialization. ''All'' denotes the whole chosen subgraph, and "Max" denotes the largest connected part of a subgraph used for property evaluation.
>
> | Run                 |          1         |         2       |           3          |           4           |         5        |
> | :----------------  | :--------------- | : -------------  | :----------  | :--------------- | :-------------  |
> | $IoU_{all}$     | 0.848+-0.163  | 0.765+-0.106 | 0.784+-0.112 | 0.829+-0.166 | 0.813+-0.186|
> | $IoU_{max}$  | 0.779+-0.330 | 0.696+-0.310 | 0.742+-0.333 | 0.757+-0.335 | 0.762+-0.304 |
>
> We have updated Table 1 in our paper.
>
> Q3: The influence of $L_{con}$ to the size of subgraph. A table showing the size of final subgraphs.
> A3: The influence of  $L_{con}$ is weak. The size of subgraph mainly depends on task relevant information.  As shown in DRD2 comparing to HLM datasets, DRD2 selects almost all the nodes because the activities of DRD2 depends on almost the whole structure of the molecules while HLM only chooses substructures of the molecules because the activities of HLM depends on some small functional groups. The weights of  $L_{con}$ are equal for two experiments, though. To clearify the influence of  $L_{con}$, we provide the size of subgraph (percent of nodes) on four datasets in Table 2 and 3, where ''All'' denotes the whole chosen subgraph, and "Max" denotes the largest connected part of a subgraph used for property evaluation.
>
> Table 2: Size of the chosen subgraphs on four datasets in percent.
>
> |Method      |        QED     |  DRD2      | HLM-CLint | MLM-CLint |
> | :----------------  | :--------------- | : -------------  | :----------  | :--------------- |
> |GCN+Att05|   43.5+-5.4  | 46.8+-3.1 | 48.1+-4.1    | 45.1+-4.1     |
> |GCN+Att07|   65.8+-3.4  | 66.7+-3.0 | 65.8+-5.7    | 67.6+-5.5     |
> |GCN+GIB   |  49.6+-15.0 | 94.8+-5.3 | 47.7+-13.7 | 54.7+-17.2   |
>
> Table 3: Size of largest connected parts used for graph interpretation in percent.
>
> |Method      |      QED       |     DRD2     | HLM-CLint | MLM-CLint|
> | :----------------  | :--------------- | : -------------  | :----------  | :--------------- |
> |GCN+Att05| 22.5+-9.5    | 34.7+-9.4   |23.5+-7.2     | 29.7+-7.9   |
> |GCN+Att07| 43.3+-11.8  | 54.2+-13.0 | 45.0+-15.2 | 41.2+-8.4    |
> |GCN+GIB   | 41.3+-16.9  | 92.8+-10.4 | 29.1+-10.6 | 36.9+-16.2  |
>
> Moreover, we provide the influence of the hyper-parameter $\alpha$ of $L_{con}$  to the size of subgraphs in Table 4. And we find its influence is limited.
>
> Table 4: The influence of the hyper-parameter $\alpha$ of $L_{con}$  to the size of subgraphs on QED dataset.
>
> |$\alpha$     |           1           |            3          |         5            |          10          |
> | :----------------  | :--------------- | : -------------  | :----------  | :--------------- |
> |All           | 0.483+-0.143 | 0.496+-0.150 | 0.494+-0.147| 0.466+-0.150 |
> |Max        | 0.387+-0.173 | 0.413+-0.169 | 0.411+-0.169| 0.391+-0.172 |
>
> We also update the tables in the paper. Please refer to Section 5.5 and Appendix A.7 for more analysis.

---

> > ### Author Response · Authors · 2020-11-18
> > **Reply (Part 2 of 2)**
> >
> > Q4: Details of datasets.
> > A4: Thanks to the reviewer for the suggestion. We update the details of the datasets in the updated version. Please refer to Appendix A.6 for more details.
> >
> > Q5: Convergence of different loss.
> > A5: We have included a plot showing the training dynamic in the revised paper. Please refer to Appendix A.8 for details.
> >
> > Q6: Larger graphs and non-uniform edge addition [optional].
> > A6: That is a good suggestion. As our model is not particularly designed for graph denoising, it is challenging to operate on larger graphs with more permuted edges. Therefore, we choose the MUTAG dataset, where the graphs are relatively small.
> > However, it is interesting to consider larger graphs with non-uniform edge addition in experiments.  And we leave them to our future work.

---

### Official Review · AnonReviewer1 · 2020-10-30
**Clearly Below Threshold**

**Rating:** 2
**Confidence:** 2

**Review:**

Authors propose to apply information bottleneck to network structured data which is represented by graphs whose nodes are assigned features.
The idea seems promising but the authors need to improve their manuscript considerably. In particular, the probabilistic model underlying the IB framework needs to be made clear right from the start. Which random graphs do you consider ?

- "... GCN outputs the node embeddings X from the following process:... " what does that mean ?
- "...the GIB seeks for the most informative yet compressed representation Z by optimizing the following objective .. " what is the domain of the optimization problem here ? and what do you mean precisely by "compresse representation"

---

> ### Author Response · Authors · 2020-11-18
> **Reply**
>
> We thank the reviewer for the comments.
>
> Q1: Which random graphs do you consider?
> A1: We do not consider random graphs. And we do not consider any priors for it.
>
> Q2: What does ‘GCN outputs the node embeddings X from the following process’ mean?
> A2: This sentence shows how graph convolutional network generally works in Notations and Preliminaries. It passes the information of neighborhood to the central node and uses an MLP to obtain the node embedding.
>
> Q3: In ‘the GIB seeks for the most informative yet compressed representation Z by optimizing the following objective’, what is the domain?
> A3: This sentence introduces a general framework of our graph information bottleneck. Here, Z is a d-dimensional vector (d is a hyper-parameter), or called the embedding of the input graph. And $G$ refers to a graph. Here we consider the subgraph recognition problem, so we replace Z with $G_{sub}$, which refers to a subgraph to define the IB-Subgraph.
>
> Q4: What do you mean precisely by ‘compressed representation’?
> A4: In the general practice of information bottleneck, a ‘compressed representation’ is a vector in the latent space that involves less dimensions than the original inputs. In subgraph recognition scenario, a ‘compressed representation’ is a subgraph which only preserves informative substructure of the graph label/property.

---

### Author Response · Authors · 2020-11-18
**Comments on a parallel work**

Differences between the Proposed Work, Graph Information Bottleneck for Subgraph Recognition (GIB-SR), and Graph Information Bottleneck (GIB):

We noticed a parallel work in NeurIPS 2020, named GIB, which was publicly available on October 24, 2020. Our GIB-SR was submitted to ICLR 2021 on October 3, 2020. These two works share similar intuitions on a high level, but have many differences, the major ones are summarized as follows:

1. Different tasks: GIB applies the information bottleneck principle to the node-level tasks, while GIB-SR aims at the graph-level tasks. That is, we argue that not all components in graph-structured data are necessary for downstream tasks, such as graph classification, graph compression and denoising. Therefore, we aim to discover a compressed yet informative subgraph (IB-Subgraph) within a graph to enhance the performance of a backbone on these tasks. In contrast, GIB learns robust node embeddings for defense against adversarial attack.

2. Substructure vs embedding: On rethinking what is a good representation of graph-structure data, GIB-SR and GIB turn to substructure (IB-Subgraph) and latent embeddings respectively. Exploring the IB-Subgraph is rather difficult as it requires our method to obtain an irregular and discrete substructure of the input, while the latent embedding is usually continuous. However, IB-Subgraph directly highlights the vital substructure for the graph label or property, which is easy to interpret and facilitate downstream tasks such as functional group discovery. On the other hand, the latent embedding is also a good representation, as shown in GIB. Moreover, as discussed in our paper, GIB-SR can be easily adapted to learn robust representations of graphs. Differently, GIB seeks for probabilistic embeddings of nodes, and thus improves the robustness of GNN for node classification.

3. Optimization strategy: On the minimization the mutual information(MI) in IB, GIB employs variational upper bounds and GIB-SR uses bilevel optimization strategy. Like the general practice in VIB, GIB introduces the non-informative prior distributions for adjacent matrix (q(A)) and featurematrix (q(Z)), thus yields the upper bounds. Furthermore, it copes with the non-i.i.d node samples as nodes are correlated.  However, as subgraphs are often irregular, it is difficult to assign a rational prior for subgraphs. Moreover, the graph samples are i.i.d. Therefore, GIB-SR adopts a bilevel optimization (see Figure 1 in our paper). That is, estimate the MI $I(G,G_{sub})$ in the inner-loop with a statistic network, which is able to cope with irregular graph data. Then, optimize the parameters of the subgraph generator to reduce $I(G,G_{sub})$.

4. Continuous relaxation: GIB-SR and GIB also differ in how they do continuous relaxation to sample the substructures and neighbors, respectively. GIB uses Categorical or Bernoulli distribution with learnable parameters to sample the neighborhood. GIB-SR learns a node assignment and uses $L_{con}$ to stabilize the training process.

---

### Comment · ~Junchi_Yu1 · 2021-06-02
**Our code is now publicly available.**

https://github.com/Samyu0304/graph-information-bottleneck-for-Subgraph-Recognition

---

### Decision · Program_Chairs · 2021-01-07
**Final Decision**

**Decision:**

Accept (Poster)

**Comment:**

This paper proposes a graph information bottleneck (GIB) framework for subgraph recognition, including the proposal of a MI objective as well as a bi-level optimization scheme for minimizing said objective. The paper receive mixed reviews, with two reviewers in favor of acceptance and two reviewers in favor of rejection.

One negative reviewer was too short to judge and had low-confidence. I think most of the concerns arise from lack of understanding of the work and the authors adequately address this on the rebuttal. The authors are encouraged to make minor modifications for clarity. In particular, classical IB considers random variables x, z, y, and learns latent representation z that is maximally informative about output y and sufficiently informative about input x. Therefore, it is natural to expect that the input to GIB is a random graph.

The other negative reviewer finds the paper lacks novelty and points to multiple references. The positive reviewers also ask about the connection with additional references. Im my opinion, the authors do an excellent job at clarifying the differences with all prior work mentioned by the reviewers, including the closest one, a GIB paper in NerurIPS 2020. In my view, the present submission contains sufficient novelty relative to prior work, specifically as it focuses on a different problem (sub-graph) and proposes a different optimization method. That being said, I think it is absolutely essential that the author responses be added to the paper. In other words, the final version must add citations to the relevant work mentioned by the reviewers and clarify the differences.

All other comments from the two remaining reviewers are very positive: the reviewers find the paper contributes with "quite interesting information theoretic objective functions that actually work on multiple graph learning tasks" and "makes a clear theoretical contribution to the field as well as provides sufficient empirical evaluation." I share the views of the positive reviewers and recommend acceptance, subject to the authors incorporating their responses to the reviewers' comments.